Towards virtual machine scheduling research based on multi-decision AHP method in the cloud computing platform

Gu Hangyu
Wang Jinjiang
Yu Junyang jyyu@henu.edu.cn
Wang Dan
Li Bohan
He Xin
Yin Xiang
Software School, Henan University , Kaifeng, Henan Province , China
Somani Arun
Electronic publication date: 2023 Nov 14
Publication date: 2023
Volume: 9
Electronic Location ID: e1675
Received 2023 Sep 17; Accepted 2023 Oct 8
Copyright: © 2023 Gu et al.
Copyright year: 2023
Copyright holder: Gu et al.
License: This is an open access article distributed under the terms of the Creative Commons Attribution License, which permits unrestricted use, distribution, reproduction and adaptation in any medium and for any purpose provided that it is properly attributed. For attribution, the original author(s), title, publication source (PeerJ Computer Science) and either DOI or URL of the article must be cited.
License URL: https://creativecommons.org/licenses/by/4.0/

Keywords: Cloud computing platforms, QoS, AHP, Virtual machine placement

Funding: Science and Technology R&D Project of Henan Province 212102210078 Key Science and Technology Project of Henan Province 201300210400 This work was supported by the Science and Technology R&D Project of Henan Province (Grant No. 212102210078); and the Key Science and Technology Project of Henan Province (Grant No. 201300210400). The funders had no role in study design, data collection and analysis, decision to publish, or preparation of the manuscript.

==============================
Virtual machine scheduling and resource allocation mechanism in the process of dynamic virtual machine consolidation is a promising access to alleviate the cloud data centers of prominent energy consumption and service level agreement violations with improvement in quality of service (QoS). In this article, we propose an efficient algorithm (AESVMP) based on the Analytic Hierarchy Process (AHP) for the virtual machine scheduling in accordance with the measure. Firstly, we take into consideration three key criteria including the host of power consumption, available resource and resource allocation balance ratio, in which the ratio can be calculated by the balance value between overall three-dimensional resource (CPU, RAM, BW) flat surface and resource allocation flat surface (when new migrated virtual machine (VM) consumed the targeted host’s resource). Then, virtual machine placement decision is determined by the application of multi-criteria decision making techniques AHP embedded with the above-mentioned three criteria. Extensive experimental results based on the CloudSim emulator using 10 PlanetLab workloads demonstrate that the proposed approach can reduce the cloud data center of number of migration, service level agreement violation (SLAV), aggregate indicators of energy comsumption (ESV) by an average of 51.76%, 67.4%, 67.6% compared with the cutting-edge method LBVMP, which validates the effectiveness.

Introduction

Cloud computing means a service model that delivers users on-demand and elastic resource requests. Users send requests for computing resources such as storage, databases, servers, applications and networks to cloud providers, where it becomes easier, cheaper and faster to access computing resources. With the application of virtualization technology, multiple cloud customers can simultaneously share physical resources, while cloud vendors create a dynamically scalable application, platform and hardware infrastructure for customers (Shu, Wang & Wang, 2014; Panda & Jana, 2019; El Mhouti, Erradi & Nasseh, 2017). As the number of cloud users proliferates and the scale of data centers increases (Bhardwaj et al., 2020; Dayarathna, Wen & Fan, 2016), it results in increasing energy consumption in cloud data centers, and it continuously increases its operating cost (Myerson, 2017). It is reported that up to now, cloud data centers have accounted for 7% (Buyya et al., 2009) of all electricity resources in the world with total operating costs at around 41% of the annual power costs of a large data center (Koomey, 2007). The energy consumed by cloud data centers is eventually emitted in the way of carbon dioxide, thus having an impact on global warming, ozone depletion and other environmental pollution. High energy consumption levels mean higher cooling requirements and costs. Another disadvantage is that the wear and tear of computing devices caused by high temperatures can affect their availability and reliability leading to serious service level agreement (SLA) violations. SLA is defined as the expected service level to the facility provider. Hence, the green development of data centers with high service quality and low energy consumption is an urgent research goal.

Dynamic virtual machine consolidation (DVMC) represents an efficient approach aimed at curtailing energy consumption while upholding quality of service (QoS). It achieves this by adjusting the workload distribution across hosts through virtual machine (VM) migration and placement. This approach tends to consolidate more VMs onto the intended host, subsequently transitioning low-workload hosts to an idle mode to curtail resource fragmentation. The overarching objectives encompass energy savings, heightened resource utilization, and an improved quality of service (QoS) within cloud data centers. Dynamic virtual machine placement also known as virtual machine dynamic scheduling, a critical component of DVMC, is to build mapping relationships with different goals. However, inappropriate virtual machine (VM) placement strategies may increase the number of additional VM migrations and have an impact on the performance of VMs and the QoS in the data center (Zhu et al., 2008; Wang et al., 2022b). Finding the optimal mapping relationship between hosts and VMs is a well-known NP-hard problem oriented towards optimization of multiple objectives (Garey & Johnson, 1983; Laghrissi & Taleb, 2018; Rozehkhani & Mahan, 2022), and a performance-guaranteed VM placement policy should meet the cloud provider’s vision of a data center with characterizes of high energy-efficiency, QoS-guaranteed and less additional VM migration.

Unlike previous articles (Tarighi, Motamedi & Sharifian, 2010; Juarez, Ejarque & Badia, 2016; Wang et al., 2022a), and also inspired by Ahmadi et al. (2022), the setting of weights for different objectives in the VM placement process is generally empirical; the Analytic Hierarchy Process (AHP) in the face of multi-objective problems implies regarding a complex multi-objective decision problem as a system. AHP decomposes the muti-objectives into several levels of multi-criteria, and calculates the hierarchical single sort and total sort by the qualitative index fuzzy quantization method, which is introduced as the systematic method of multi-objective optimization decision. Therefore, AHP shows more rational, scientific, and decisional intelligence compared with the drawbacks.

In this article, we propose a novel AHP decision-based virtual machine placement policy (AESVMP), where the decision criteria consist of resource allocation balance rate, host power consumption and available resources. The resource allocation balance rate is derived from our proposed balance-aware resource allocation function, which calculates the parallelism between the total resource plat surface of the host in three dimensions and the allocated resource plat surface (CPU, RAM, BW); secondly, the increase in power consumption of the host after placement and thirdly the available resources of the host. Then, they serve as the decision criteria for evaluating the targeted hosts for VM migration. Eventually, this article leverages the AHP to calculate the scientific weights of the above decision criteria for seeking for the appropriate host. The main contributions of this article are as follows: AHP-based resource balance-aware and energy-optimized virtual machine placement policy (AESVMP) is introduced to tackle the dynamic VMP problem.

A standard function for balanced resource allocation is introduced to achieve balanced resource utilization of hosts.

Simulations using real-world workloads PlanetLab on CloudSim demonstrated improvements in energy consumption, number of VM migrations and service level agreement violation (SLAV) compared with current state-of-the-art VM scheduling strategy.

The remaining sections of this article are organized as follows. Related work is discussed in “Related Work”. “Virtual Machine Placement Policy Based on AHP Resource Balancing Allocation” presents a virtual machine placement policy based on AHP resource balancing allocation. “Experimental Evaluation” evaluates the proposed approach based on the experimental environment, performance metrics, comparative benchmarks and illustrates extensive simulation results. “Conclusion” concludes and describes future work.

Related works

Virtualization is the core technology for cloud environment. How to find the most proper targeted host for migrated VM with assistance of the technology is also research direction. Masdari, Nabavi & Ahmadi (2016) and Talebian et al. (2020) introduce exhaustively the development of virtual machine placement. Table 1 gives a relevant features comparison of related works.

Table 1 Summary of related works techniques.

Methods	Strengthens	Weakness	
LBVMP (Wang et al., 2022b)	Considers the ratio of the PM’s available resources (CPU, RAM, BW) to the VM’s requested resources as a reference standard	Ignore other criteria such as power consumption and weighting of individual indicators	
PABFD (Beloglazov & Buyya, 2012)	Select the host with the least increase in power consumption after placement	Ignore resources contention and resources balance	
HPSOLF-FPO (Mejahed & Elshrkawey, 2022)	Multi-objective decision making to optimize power consumption and resource utilization	Ignore resources balance	
SAI-GA (Karthikeyan, 2023)	Selects the best host based on CPU and memory usage using an adaptive genetic algorithm	Ignore bandwidth cost and resources balance	
GMPR (Wang et al., 2023)	Further optimization of PM resource utilization and energy consumption by considering resource wastage	Ignore power consumption and resources balance	
VMP-A3C (Wei et al., 2023)	Use deep reinforcement learning to maximise load balancing and minimise energy consumption	Ignore bandwidth cost	
PRUVMS (Garg, Singh & Goraya, 2022)	Use resource utilization and power consumption as criteria for selecting the right PM	Ignore resources balance and SLA violation	
MRAT-MACO (Nikzad, Barzegar & Motameni, 2022)	Finding optimal VM placement solutions using SLA-aware multi-objective ant colony algorithm	Ignore bandwidth cost and resources balance	
CUECC (Wang et al., 2022a)	Improve service quality by judgement host cpu utilization and power consumption	Ignore resources balance and bandwidth cost	
MEEVMP (Sunil & Patel, 2023)	SLA violation, energy usage and power efficiency of PM are taken into account in the VM placement	Ignore bandwidth cost and resources balance	
VM-DFS (Zhuo et al., 2014)	Reduce the number of active hosts by predicting memory requirements for the next service cycle	Ignore cpu cost and bandwidth cost	
VMCUP-M (Hieu, Francesco & Ylä-Jääski, 2020)	Predict resource utilization of hosts for the next service cycle, reducing the number of VM migrations	Ignore resources balance and power consumption	
MOPFGA (Liu et al., 2023)	Heat recirculation around the PM rack is used as a reference criterion for selecting the host	Ignore bandwidth cost and resources balance	
priority-aware (Omer et al., 2021)	Further optimize energy consumption and resource utilization by selecting the right PM through traffic priority and power consumption	Ignore resources balance	
VMDPA (Chang et al., 2022)	Choose a host with faster data transfer speeds and lower bandwidth costs	Ignore energy consumption and resources balance	
KCS (Mukhija & Sachdeva, 2023)	Integrates bio inspired cuckoo search with unsupervised K clustering machine learning algorithm	Ignore bandwidth cost and resources balance	
Q-learning (Aghasi et al., 2023)	Use a decentralized Q-learning approach to accomplish the Energy-efficient and thermal-aware placement of virtual machines	Ignore resources balance	
AGM-VMP (Li, Pan & Yu, 2022)	PMs with more available resources are given higher priority, further reducing the probability of PM overloading	Ignore energy consumption and weighting of individual indicators	
CUECC (Wang et al., 2022a)	Selection of appropriate PM based on predicted host CPU resource utilization	Ignore bandwidth cost and resources balance	
VM-DFS (Zhuo et al., 2014)	Predict VM memory requirements for the next service cycle and using the boxing algorithm	Ignore bandwidth cost and resources balance	
VMCUP-M (Hieu, Francesco & Ylä-Jääski, 2020)	Predict host resource utilization for the next service cycle	Ignore resources balance	

Beloglazov & Buyya (2012) propose a power aware best fit decreasing (PABFD) algorithm for VM placement. The PABFD algorithm sorts the to-be-migrated VMs in the descending order of CPU resource utilization. Then, it sequentially seeks destination hosts for the VMs while guaranteeing minimum energy consumption of the destination hosts after VM placement. Load balancing plays a pivotal role in VM placement. Uneven allocation of host resources can result in suboptimal resource utilization, performance deterioration, and consequently, a diminished quality of service. Load balancing is referenced in several of the following VM placement policies. Wang et al. (2022b) suggest a VM placement strategy called LBVMP. Define the two planes to be the available resource plane of the PM (CPU, RAM, BW) and the resource plane required by the VM respectively. LBVMP then calculates the distance between the two plats to evaluate the VM allocation solution. Karthikeyan (2023) devise a genetic algorithm to decide the best matching host based on CPU and memory usage. Wei et al. (2023) introduce deep reinforcement learning (DRL)-based strategies to enhance load balancing, aiming to ascertain the optimal mapping between VMs and PMs. Li, Pan & Yu (2022) design a virtual machine placement strategy based on multi-resource co-optimization control. Assessing the likelihood of a well-balanced multi-resource utilization status of a PM through Gaussian distribution estimation. Mejahed & Elshrkawey (2022) provide a multi-objective decision making approach which considers placement time, energy consumption and resource wastage separately. Energy consumption and resource utilization are further optimized.

Most other VM placement strategies don’t take load balancing into account. Nikzad, Barzegar & Motameni (2022) suggest a multi-objective search for optimal virtual machine placement solutions based on ant colony algorithm. Mukhija & Sachdeva (2023) propose a KCS algorithm. It integrates bio inspired cuckoo search with unsupervised K clustering machine learning algorithm for resolving the VMP problem. Wang et al. (2023) devise a two-phase greedy virtual machine placement algorithm to further reduce energy consumption and resource wastage. Garg, Singh & Goraya (2022) devise an energy consumption and resource utilization aware virtual machine scheduling algorithm (PRUVMS) to effectively reduce energy consumption and the number of virtual machine migrations and improve resource utilization. Sunil & Patel (2023) design a virtual machine placement strategy based on a packing algorithm that reduces the overall energy consumption of a data centre. Liu et al. (2023) suggest a novel thermal-aware VM placement strategy to solve the problem by jointly considering energy consumption and heat recirculation around PM racks. Chang et al. (2022) design a optimized VM placement algorithm considering data transfer velocity, cloud storage performance, and network bandwidth. Aghasi et al. (2023) provide a Q-learning based VM placement strategy. Optimize energy consumption and keep the host temperature as low as possible while satisfying service level agreements (SLA). Omer et al. (2021) propose a VM placement strategy with consideration of both energy consumption and traffic priority. For critical applications, select energy-saving PM. For normal type, select sufficient resources PM. The suggestion enables to reduce energy consumption and resource wastage.

The application load submitted by users to the data center is dynamically variable, so the resource utilization of the hosts in the data center also fluctuates over time (Hieu, Francesco & Ylä-Jääski, 2020). The methodology of prediction can predict the future workload conditions of hosts, VMs. Zhuo et al. (2014) propose a VM dynamic predictive scheduling algorithm (VM-DFS). Selecting a PM that meets predicted memory requirements. The number of active hosts is reduced to ensure that the resource requirements of the VMs are satisfied. Wang et al. (2022a) propose a host state detection algorithm based on a combination of grey and ARIMA model. In addition, they propose a CPU utilization and energy-aware VM placement strategy based on the prediction results. Hieu, Francesco & Ylä-Jääski (2020) propose a multi-purpose predictive virtual machine integration algorithm (VMCUP-M). The future utilization of a variety of resources is predicted by using the historical data of the host, and the results of multiple predictions of multiple resources are applied in the process of VM migration selection and targeted host placement, which effectively improves the performance of the cloud data center.

The algorithm proposed based on the application of AHP in this article takes three key criteria into account the power consumption increase, available resources and resource allocation balance ratio of the host respectively. In addition to reducing energy consumption, SLAV ensures the quality of service (Beloglazov & Buyya, 2017; Gmach et al., 2009) and performance-guaranteed for the cloud data center.

Virtual machine placement policy based on ahp resource balancing allocation

We first formulate the energy consumption and the overhead of VM migration and briefly give an introduction of the resources allocation balance ratio. Subsequently, we present a VM placement policy computed using analytic hierarchy process(AHP) based on the resource allocation balance ratio and two other criteria and illustrate the proposed algorithms in the end. Table 2 lists the symbols used for the readability of the rest of the article.

Table 2 List of notations used in this article.

Notation	Description	
AHP	Analytic hierarchy process	
H	A set of physical machines; H = {h1,h2,…,hM}	
V	A set of virtual machines; V={v1,v2,…,vN}	
P(ui)	The power consumption of hi	
Ui(T)	The CPU utilization of hi at time T	
Pimax	The max power of hi	
Piidle	The power of hi with idle state	
Ui	The CPU utilization of hi	
Ei	Energy consumption generated by the hi	
Etotal	The total energy consumption of all active hosts in the cloud data center	
Uj(t)	The CPU utilization of vj at time t	
PFjdegradation	Migration cost of vj	
vjRam	The RAM resource request capacity of vj	
hiBW	The available BW resource capacity of hi	
tjmig	Migration time of vj	
Δhostiallocated	The allocated resources flat surface of host hi	
Δhostitotal	The total resources flat surface of host hi	
CPUiallocated	Host hi Already allocated CPU resources	
RAMiallocated	Host hi Already allocated RAM resources	
BWiallocated	Host hi Already allocated BW resources	
CPUitotal	Total CPU resources of host hi	
RAMitotal	Total RAM resources of host hi	
BWitotal	Total BW resources of host hi	
Normaliallocated→	The normal vector of Δhostiallocated	
Normalitotal→	The normal vector of Δhostitotal	
Balanceiser	The resources allocation balance ratio of host hi	
Norpoweri	The function power consumption of host hi.	
ARi	Available resource of host hi.	
Uiincre	Increase in CPU utilization of host hi after placement	
Xij	Mapping of host hi to vm vj	
vjmips	The cpu resource requirements of vm vj	
himips	The cpu resources of host hi	
W¯	Weighting matrix for the three decision-making criteria ( Balanceiser, Norpoweri, ARi)	
W	Normalisation of the judgement matrix ( W¯i)	
λmax	Maximum characteristic root	
C.I.	Consistency Index	
R.I.	Stochastic Consistency Indicator from 1,000 Satty Simulations	
C.R.	For degree of conformance validation	
scoreHosti	Host scores obtained through the methods in this article.	

Model of energy consumption and overhead of virtual machine migration

In a cloud data center, memory, CPU, cooling systems and other devices all have significant energy demands. Energy consumption of the CPU accounts for approximately 61% of the total power consumed in the data center (Mejahed & Elshrkawey, 2022), so the energy consumption and power consumption of the hosts in the data center varies with CPU utilization (Wang et al., 2022b), the power consumption of the host P(Ui) is derived from Eq. (1). (1) P(Ui)=Piidle+(Pimax−Piidle)×Ui

where Pimax, Piidle, Ui present maximum power of host when experiencing full CPU utilization, minimum power of host with sleep state and host’s CPU utilization respectively. Energy consumption of host (Ei) came from Eq. (2), thus the total energy consumption of all active hosts in the cloud data center (Etotal) stems from Eq. (3). (2) Ei=∫T1T2P(Ui(T))dT

(3) Etotal=∑i=1MEi

When the VM migration module is triggered. The average performance degradation of the VMs affected by migration is roughly 10% of the CPU utilization of the VMs (Wang et al., 2022b). Therefore, the overhead of VM migration PFjdegradation is defined as follows: (4) PFjdegradation=110×∫t0t0+tjmigUj(t)dt

(5) tjmig=vjRamhiBW

where the tjmig is calculated by Eq. (5). Where, Uj (t) presents the CPU utilization of VM vi at time t, vjRam expresses the RAM resource request capacity of vi and hiBW represents the available BW resource capacity of host hi.

The formulation of balanced resource allocation

Unbalanced resource allocation of hosts means that hosts with high resource utilization in single or multiple resource dimensions, and the available resources are not sufficient to allocate to the migrated VMs to ensure their working, which eventually leads to resource loss of the hosts. Thus, a balanced allocation of host resources can further improve the efficiency of resource allocation in cloud data centers and guarantee their quality of service.

CPU utilization accounts for a large proportion of hosts’ energy consumption, and RAM and BW are closely bound up with service level agreement violation (SLAV). Hence, the physical resources (CPU, RAM, BW) utilization reflects (Wang et al., 2022b) the impact on working performance of VMs to some extent. To evaluate the available resource parallelism that targeted host allocates resources for migrated VM with the account of CPU, RAM, BW (Ferdaus et al., 2014). In 3-D resource as shown Fig.1, the allocated resources flat surface of host hi according to real-time resources utilization, Δhostiallocated, is determined and denoted by Eq. (6), while the total resources flat surface of host hi, Δhostitotal, is presented by Eq. (7) respectively.

Figure 1 The allocated resource flat surface Δhostallocated when VM placed on host and host’s total resource flat surface Δhosttotal.

(6) Δhostiallocated:1=CPUiallocatedCPU+RAMiallocatedRAM+BWiallocatedBW

(7) Δhostitotal:1=CPUitotalCPU+RAMitotalRAM+BWitotalBW

If the angle between the host’s allocated resource flat surface and the host’s overall resource flat surface is smaller and closer to parallel, it means that the host’s resources are more equally allocated. Thus, the resources allocation balance ratio denoted by Balanceiser is defined as the cosine value between flat surface Δhostitotal and flat surface Δhostiallocated. The balance of resource allocation for hosti is inversely proportional to the value of Balanceiser, with a smaller value of Balanceiser indicating a more balanced allocation of the host.

Now we give the solution to the computation of normal vector of flat surface Δhostitotal and Δhostiallocated defined as Eqs. (8) and (9). We assume that Normalitotal→ and Normaliallocated→ denoted by normal vector of flat surface Δhostitotal and Δhostiallocated respectively, and the value Balanceiser is calculated below: (8) Normalitotal→=(BWitotal−CPUitotal)∗(RAMitotal−CPUitotal)

(9) Normaliallocated→=(BWiallocated−CPUiallocated)∗(RAMiallocated−CPUiallocated)

(10) Balanceiser=cos<Normalitotal→,Normaliallocated→>=(Normalitotal→×Normaliallocated→)(∣Normalitotal→×Normaliallocated→∣)

The proposed approach

We assume that the data center contains a set of heterogeneous hosts H={h1,h2,…,hM} ( i∈<1,…,M>) and a set of VMs V={v1,v2,…,vN} ( j∈<1,…,N>) and each host also hosts multiple VMs, intuitively as shown in Fig. 2. In this article, we mainly take into consideration resource type CPU, RAM, BW. When a user submits a resource request to the cloud provider, the cloud data center will provide a real-time service to create the VM instance, which will consume the resources of the physical machine in terms of CPU, RAM and BW. A host exhibiting high resource utilization can impact the performance of the VM. This is because the running VMs co-compete for host resources to fulfill their variable workload demands. When the VMM manager module is triggered and communicates with the VMP manager, the VMP manager establishes a more appropriate mapping relationship between VMs and hosts. This is achieved by employing an AHP-based resource balance-aware VMP strategy. The overarching goals encompass minimizing energy consumption, reducing the number of additional VM migrations, and alleviating service level agreement violations within a cloud data center.

Figure 2 The architecture framework.

AHP-based virtual machine placement strategy

The analytic hierarchy process (AHP), one of the multi-attribute decision-making models (Saaty, 1990), is to decompose a complicated problem into a number of levels and a number of influential factors, then to hierarchize the influential factors and transfer factors in data-form. It uses mathematical methods to calculate the relative weights of a number of influences affecting the decision. Ultimately find the best solution to the problem. The overall process is to first identify the criteria that influence the decision and construct a hierarchical decision tree. Subsequently a judgement matrix is developed based on the decision objectives. Then calculate the relative weights of each criterion, and obtain the weight matrix of each criterion after passing the consistency test. A analytic hierarchy process (AHP) is used to solve the VM placement problem. Decision criteria affecting VM placement and their judgement matrices are first identified. Then the relative weights of these decision criteria are calculated and the optimal host is found for the migrated VMs based on the weights. The detailed steps are as follows.

Step 1: determining the decision criteria and the hierarchical decision tree

Firstly, a hierarchical model is constructed. The first targeted layer is to dynamic virtual machine placement with energy savings and QoS guarantees, it alleviates energy consumption and SLAV for cloud data centers in the execution of VM scheduling. The second layer denotes the decision layer with three main criteria: power consumption, resource allocation balance ratio and available resources, and the third layer represents the available physical hosts ( hi). The decision tree composed of these three layers is shown in Fig. 3. When VMM communicates with the VMP manager, the VMP manager module executes scheduling with the following three criteria: the increased power consumption ( Norpoweri) of host hi.

resources allocation balance ratio ( Balanceiser) of host hi.

available resource ( ARi) of host hi.

Figure 3 Hierarchy model.

In the virtual machine placement process, the AHP-based decision criteria include power consumption, available resources, and resource allocation balance ratio. This framework governs the dynamic execution of virtual machine placement (VMP) with the objectives of minimizing energy consumption, reducing the number of VM migrations, and mitigating the impact of additional VM migrations on SLA violations that may result in performance degradation.

We use Xij to represent the mapping relationship between the VM and the host, which is defined as Eq. (11). When Xij is 1 it means that VM vj is placed on host hi. (11) Xij= {1′if  vj  place  on  hi0′ if  otherwise

Upon placing the migrated VM onto the designated host, there is a subsequent rise in both CPU utilization and power consumption. The elevated CPU utilization, along with the resulting fluctuations in power consumption, are represented as uiincre and powerDiffi, respectively. powerDiffi is computed using the Eq. (12). To further unify the calculation normalize powerDiffi to Norpoweri using the Eq. (13). Norpoweri as one of the three main decision criteria. (12) powerDiffi=P(Ui+Uiincre)−P(Ui)

(13) Norpoweri=(1−11+e−powerDiffi)

A host with more available resources provides performance guarantees for virtual machines and also reduces the number of virtual machine migrations. In comparison to the energy consumption generated by RAM and BW resources, CPU resources account for the largest proportion of energy consumption in cloud data centers (61%) (Kusic et al., 2009). Therefore the amount of CPU available resources ( ARi) is used as one of the three main decision criteria in the VM placement process. ARi is calculated as Eq. (14). himips and vjmips represent the total CPU resources of host hi and the CPU resource requirements of the vj, respectively. (14) ARi=1−∑j=1NXij×uj×vjmipshimips

Step 2: determining the weight of the criteria concerning the goal

The second step after determining the criteria is to form the judgment matrix of criteria based on the model’s objective and determine the priority of the criteria. The construction of a 3 * 3 judgment matrix A using the three criteria above is shown in Table 3, where the element Apq (p,q∈[1,3]) indicates the importance of the value in row p compared to it in column q. Meanwhile, this matrix is given as input to Algorithm 1.

Table 3 Judgment matrix(A).

	Norpoweri	Balanceiser	ARi	
Norpoweri	1	2	12	
Balanceiser	12	1	15	
ARi	2	5	1	

Algorithm 1 Calculate weight.

  Input: Judgment matrix A, int n // n: number of dimensions of the matrix	
  Output: The weight matrix W	
1   x∈<0,n−1>, y∈<0,n−1>;	
2  Double [n][1] W;	
3  Double [n][1] W¯;	
4  Double sum = 0;	
5  Double λmax = 0;	
6  for int x = 0; x < n; x++ do	
7    W¯[x][0]=∏y=0n−1A[x][y]n−1; // Calculate the weighting matrix Eq. (15)	
8  end	
9  for int x = 0; x < n; x++ do	
10    W[x][0]=W¯[x][0]∑y=0n−1W¯[y][0] ; // Normalizing the weighting Eq. (16)	
11  end	
12  for int x = 0; x < n; x++ do	
13   Doublesum=∑y=0n−1A∗W[x][y]W[x][0]+sum; // Calculate the value of the product matrix of the judgement matrix and the weight matrix	
14  end	
15  λmax = sum / n; // Calculate the maximum characteristic root	
16 Double C.I. = ( λmax−n)/(n−1);	
17 Double C.R. = C.I./R.I.;	
18  if C.R.>= 0.1 then	
19  return("Unreasonable weighting"); // Failed consistency test	
20  end	
21  if C.R.< 0.1 then	
22   return(W); // Passes consistency test	
23  end	

Then stratified single sort and consistency tests are performed separately. The weight matrix of the three decision criteria ( W¯) is calculated by the Eq. (15). The standardised matrix (W) is calculated using the Eq. (16) as shown in Table 4.

Table 4 Criteria weight.

Norpoweri	Balanceiser	ARi	
0.280	0.131	0.589	

(15) W¯=∏q=13Apq3p,q=1,2,3

(16) W=W¯∑p=13W¯

Subsequent consistency test. The judgement matrix A and the weight matrix W are used to calculate the maximum characteristic root according to Eq. (17). Then calculate C.I. and C.R. from Eqs. (18) and (19). Eventually, the outcome is shown in Table 5. It can be seen that the value C.R. is equal to 0.0051 and less than R.I., which demonstrates that the criteria weight matrix passed the test. Thus hosts can be selected based on the weight matrix W during VM placement.

Table 5 λmax, C.I., C.R., R.I. calculation results.

λmax	C.I.	C.R.	R.I.	
3.006	0.003	0.0051	0.58	

(17) λmax=13∑p=13(AW)W

(18) C.I.=λmax−33−1

(19) C.R.=C.I.R.I.

Step 3: calculation of host score based on criteria

Since the purpose of using the AHP method in this article is to determine the relative weights between the three decision criteria, a hierarchical total ranking is not required. (20) scoreHosti=[NorpoweriBalanceiserARi]×W

Ultimately, from the above two steps, the relative weight matrix (W) of the three decision criteria can be obtained. The three standard indices of available hosts are multiplied with the weight matrix to obtain the host’s score ( scoreHost) by Eq. (20). The host score is calculated using Eq. (20), and the host with the highest score is considered the most suitable host for placement. One of the strengths of the proposed approach is the flexibility it offers (Ahmadi et al., 2022), the relative weights calculated can be flexibly altered by data centers’ preferences.

AESVMP algorithm

The monitoring procedures of the data center periodically monitor the working status of the servers. A host with high resource utilization will suffer from resource contention among the VMs working on it, resulting in performance degradation. Therefore, the management system triggers VM migration according to the AESVMP mechanism proposed in this article, as shown in Algorithm 2, to build a new mapping relationship between migrated VM and host in the cloud data center with the goals of energy saving, reduction of SLAV and the number of VMs migration.

Algorithm 2 AESVMP.

  Input: HostList, VmsToMigrate // HostList: the list of host; vmsToMigrate: some VMs prepare to migrate	
  Output: migrationMap // gaining the mapping relationship between host and VM	
1  Initialize migrationMap = List<Map<String, Object>>;	
2  Initialize allocatedHost = null;	
3  Double minCriterion = Minvalue;	
4  List switchedOffHosts = getSwitchedOffHosts();	
5  sortByCpuUtilization(vmsToMigrate);	
6  for VM vm: VmsTomigrate do	
7   for Host host: HostList do	
8    if host is overloadedhost then	
9     continue; // Exclude overloaded hosts	
10   end	
11   if host is switchedOffHosts then	
12    continue; // Exclude sleep state hosts	
13   end	
14   if host.isSuitableForVm(vm) then	
15     scoreHosti=[NorpoweriBalanceiserARi]×W; // The three decision criteria are multiplied by the corresponding weights to obtain the host score. Where W is the weight matrix obtained by Algorithm 1	
16    if scoreHosti> minCriterion then	
17     minCriterion = scoreHosti	
18     AllocatedHost = host; // Choose the host with the highest score based on three criteria	
19    end	
20   end	
21  end	
22  Initialize migrate = Map < String, Object >;	
23  migrate. put (“vm”, vm);	
24  migrate. put (“host”, allocatedHost);	
25  migrationMap. add(migrate);	
26 end	
27 return migrationMap;	

The application of Algorithm 1 is to calculate the weigh matrix of the decision criteria using AHP method when VM explores the targeted host. First, the judgment matrix of decision criteria (A) and the number of dimensions of the matrix ( n) are used as the input of the algorithm. We define various variables (lines 2–5). The weights of each decision criterion are calculated based on the judgment matrix to obtain the weight matrix ( W¯) (lines 7). The normalizing weight matrix (W) is obtained through Eq. (15) (lines 10). Then λmax, C.I. and C.R. are then calculated and used for the consistency test (lines 12–17). Output the standard weight matrix (W) after the final consistency test is passed (lines 18–23).

Algorithm 2 (AESVMP) illustrates the process of VM placement based on the Algorithm 1. First, the inputs to the algorithm are a list of migrated VMs and a list of hosts. Exclude hosts that are overloaded and dormant in (lines 8–13). The next step continues only if the condition is met that the available resources of the host exceed the requested resources of the migrated VM (line 14). Then, find the targeted host with the maximum score according to the calculation of Eqs. (10)–(20) (lines 15–19) for the migrated VM. Finally, return the result of the mapping relationship between VMs and hosts.

Time complexity analysis: We assume that the number of N migrated VMs and a set of M hosts are selected, the time complexity of performing a descending sort is O(MlogM). When triggering VMP, it is clear that the time complexity of targeted host is O(N), so the time complexity of Algorithm 2 is O(MlogM+NM), and in the worst case when M equals N, the time complexity is O(N2).

Experimental evaluation

In this section, we introduce the experimental environment, evaluation metrics and comparison benchmarks to validate the performance of the proposed approach.

Experimental environment

The proposed approach is to validate its performance under CloudSim emulator (Calheiros et al., 2011) in this article. In the tool, we simulate 800 heterogeneous host with two types HP ProLiant ML110G4 (Intel Xeon 3040) and HP ProLiant ML110G5 (Intel Xeon 3075) equally. The two types with same characterize of the number of CPU cores, RAM, BW and storage but are different in CPU capacity with 1860 MIPS and 2660 MIPS, respectively. The relationship between energy consumption and CPU utilization of the host is shown in Table 6. Then, four types of Amazon EC2 VMs, specific information as shown in Table 7, and PlanetLab project with 10 workloads, specific information as shown in Table 8, are taken into consideration during the experiment.

Table 6 Power consumption of the servers at different load levels (in Watts).

Host type	0%	10%	20%	30%	40%	50%	60%	70%	80%	90%	100%	
G4	86	89.4	92.6	96	99.5	102	106	108	112	114	117	
G5	93.7	97	101	105	110	116	121	125	129	133	135	

Table 7 Configurations for Amazon EC2 VMs.

VM type	CPU(MIPS)	RAM (GB)	Number for cores	Network BandWidth	
High-CPU medium instance	2,500	0.85	1	100 Mbit/s	
Extra-large instance	2,000	1.7	1	100 Mbit/s	
Small instance	1,000	1.7	1	100 Mbit/s	
Micro instance	500	0.613	1	100 Mbit/s	

Table 8 Planetlab trace data.

Workloads	Date	Number of servers	Number of VMs	Mean	St.dev	
w1	2011/03/03	800	1,052	12.31%	17.09%	
w2	2011/03/06	800	898	11.4%	16.83%	
w3	2011/03/09	800	1,061	10.70%	15.57%	
w4	2011/03/22	800	1,516	9.26%	12.78%	
w5	2011/03/25	800	1,078	10.56%	14.14%	
w6	2011/04/03	800	1,463	12.39%	16.55%	
w7	2011/04/09	800	1,358	11.12%	15.09%	
w8	2011/04/11	800	1,233	11.56%	15.07%	
w9	2011/04/12	800	1,054	11.54%	15.15%	
w10	2011/04/20	800	1,033	10.43%	15.21%	

Evaluation metrics

For the experimental results, the following mainstream performance indicators (energy consumption, the number of virtual machine migration, SLA time per active host (SLTAH), perf degradation due to migration (PDM), service level agreement violation (SLAV) and aggregate indicators of energy consumption (ESV)) are determined to evaluate the performance of the proposed algorithm. These performance indicators are described below:

1. Where energy consumption represents the total energy consumption generated (Garg, Singh & Goraya, 2018) by all hosts running simulated workloads in the cloud data centers.

2. The number of VM migrations means the total number of VM migrations performed during the experiment. If the data center detects a host with a overload or underload state, then it starts VM migration. VM migration affects the performance of VM workloads. Therefore, the fewer VMs migrated, the better.

3. The working performance of migrated VM will be affected due to VM migration technology triggering, thus the performance degradation denoted by PDM is defined below: (21) PDM=1N∑i=1NPFjdegradationCjdemand

where N, PFjdegradation and Cjdemand presents the number of VM, performance degradation and total CPU capacity of vmj, respectively.

4. The user submits a request to create a VM instance to the cloud data center and signs a service level agreement with the cloud vendors. As defined in Beloglazov & Buyya (2012), service level agreements refer to the ability of the host and the previously recommended software measurement environment to meet the business quality requirements. SLA time per active host (SLATAH) indicates the percentage that the time of host with 100% CPU utilization divides the time of an active host. (22) SLATAH=1M∑i=1MTioverTiactive

where M, Tiover and Tiactive presents the number of active host, the time of host experiencing 100% CPU utilization and the time of active host, respectively.

5. SLAV is a indicator, service level agreement, to evaluate the overloaded host and performance degradation in combination with SLATAH and PDM. (23) SLAV=SLATAH×PDM

6. ESV is an metric in association with total energy consumption ( Etotal) and service level agreement violations (SLAV) and is calculated below: (24) ESV=E×SLAV

Comparison benchmarks

To validate the efficacy of the method proposed in this article, we employed five distinct host state detection methods (THR, IQR, LR, MAD, LRR) and two VM migration techniques (minimum migration time—MMT, maximum correlation—MC) within CloudSim. These were utilized to conduct a comprehensive comparative analysis of the experimental outcomes involving the AESVMP algorithm, the PABFD algorithm (Beloglazov & Buyya, 2012), and the LBVMP algorithm (Wang et al., 2022b). For the sake of comparison, we computed the average results obtained from the five distinct host state detection methods (THR, IQR, LR, MAD, LRR). The safety parameter was set to 1.2 for IQR, LR, LRR, and MAD, while for THR, it was set to 0.8. All the comparative experiments were conducted using CloudSim, with a workload derived from 10 PlanetLab instances.

Experimental results

In this section, 10 workloads (Park & Pai, 2006; Chun et al., 2003) and the performance metrics mentioned are used evaluate the performance of the proposed AESVMP algorithm compared with the VM placement algorithm discussed above.

Table 9 shows the simulation results of the performance comparison between the AESVMP algorithm proposed in this article and the state-of-the-art LBVMP algorithm (Wang et al., 2022b) with the same condition. Where compared with the LBVMP algorithm the AESVMP algorithm outperforms in terms of the number of VM migrations, SLAV, ESV and Energy efficiency, with an average optimization of 51.76%, 67.4%, and 67.6% respectively, but AESVMP performs worse than LBVMP when it comes to energy consumption. We can conclude that the approach effectively optimizes in number of VM migration and the QoS.

Table 9 Comparison of AESVMP and LBVMP algorithm.

Type	Algorithms	Migrations	SLAV ( 10−4)	Energy (kWh)	ESV ( 10−1)	
AESVMP	iqr_mmt	10,625	0.672	134.19	0.088	
	lr_mmt	10,881	0.815	145.27	0.11	
	lrr_mmt	10,881	0.815	145.27	0.11	
	mad_mmt	4,131	0.128	128.79	0.015	
	thr_mmt	7,428	0.383	131.28	0.049	
LBVMP	iqr_mmt	22,587	2.86	119.86	0.34	
	lr_mmt	15,148	1.34	145.75	0.19	
	lrr_mmt	15,148	1.34	145.75	0.19	
	mad_mmt	22,813	2.69	103.57	0.28	
	thr_mmt	23,191	2.98	114.19	0.34	

Evaluation based on energy consumption

Figure 4 shows the total energy consumption generated by different methods in combination with two VM selection algorithms MMT and MC. When the VM selection methods are MMT and MC, the average energy consumption of AESVMP strategy is reduced by 27.9% and 27.7% compared to PABFD strategy, respectively. AESVMP takes into consideration criteria Norpoweri and Balanceiser to select the host with high energy-efficiency and underlines resource allocation balance, which can reduce energy consumption. However, the AESVMP strategy has a slightly higher energy consumption than the LBVMP strategy. This may be due to the fact that AESVMP, when selecting hosts with the same conditions, prioritizes hosts with more available resources and a more balanced resource allocation. It focuses on meeting the resource requirements of VMs to improve the quality of service. As a result, the data center employing the AESVMP strategy has a higher number of active hosts compared to the one using the LBVMP strategy, leading to a slight increase in energy consumption. While LBVMP is more focused on energy consumption optimization

Figure 4 Comparison of energy consumption.

Evaluation based on number of migrations

Figure 5 depicts a comparison of the performance metrics (the number of VM migrations) for the PABFD, LBVMP, and AESVMP strategies. When comparing the AESVMP strategy to the PABFD strategy, there is an average reduction of 68.5% and 73.1% when using the VM selection algorithms MMT and MC, respectively. Similarly, when comparing the AESVMP strategy to the LBVMP strategy, there is an average reduction of 57% and 52.7% when using the VM selection algorithms MMT and MC, respectively. The AESVMP strategy considers the available resource criteria of the host and ensures the fulfillment of resource requests from virtual machines. Consequently, this approach proves effective in reducing the number of additional VM migrations.

Figure 5 Comparison of number of migrations.

Evaluation based on PDM

Figure 6 depicts a comparison of performance metrics (PDM) for the PABFD, LBVMP, and AESVMP strategies. When comparing the AESVMP strategy to the PABFD strategy, there is an average reduction of 74.4% and 73.1% when using the VM selection algorithms MMT and MC, respectively. Similarly, when comparing the AESVMP strategy to the LBVMP strategy, there is an average reduction of 52.3% and 53.7% when using the VM selection algorithms MMT and MC, respectively. This reduction in the number of migrations directly contributes to the decline in PDM values. Consequently, it highlights the effectiveness of the AESVMP strategy in optimizing both the number of migrations and the PDM metric.

Figure 6 Comparison of PDM.

Evaluation based on SLATAH

Figure 7 demonstrates variations in the SLATAH performance metric using different methods. In an intuitive comparison, the AESVMP strategy, in contrast to the PABFD strategy, achieves an average reduction in SLATAH of 73.5% and 67.1% when the VM selection methods are MMT and MC, respectively. Similarly, when comparing the AESVMP strategy to the LBVMP strategy, there is an average reduction of 49.6% and 54.7% when using the VM selection algorithms MMT and MC, respectively. The AESVMP strategy prioritizes the criteria of available resources and resource allocation balance ratio. By selecting hosts with sufficient resource capacity and emphasizing balanced resource allocation, this approach reduces the likelihood of hosts becoming overloaded. Consequently, the SLATAH metric experiences a notable decline.

Figure 7 Comparison of SLATAH.

Evaluation based on SLAV

Figure 8 illustrates the SLAV performance metrics for various methods. When comparing the AESVMP strategy to the PABFD strategy, there is an average reduction of 94.2% and 90.3% when using the VM selection algorithms MMT and MC, respectively. Similarly, when comparing the AESVMP strategy to the LBVMP strategy, there is an average reduction of 74.1% and 77.4% when using the VM selection algorithms MMT and MC, respectively. This optimization of SLAV is directly linked to the reduction in both PDM and SLATAH. Thus the AESVMP strategy significantly reduces SLA violations.

Figure 8 Comparison of SLAV.

Evaluation based on ESV

Figure 9 illustrates the performance metric ESV in relation to different methods. Intuitively, when compared to the PABFD strategy, the AESVMP strategy results in a reduction of ESV by an average of 95.7% and 92.8% when the VM selection methods are MMT and MC, respectively. Similarly, when compared to the LBVMP strategy, the AESVMP strategy reduces ESV by an average of 69.7% and 73.2% with VM selection methods such as MMT and MC. This substantial reduction in ESV is closely tied to energy consumption and SLAV.

Figure 9 Comparison of ESV.

Conclusion

In this article, we propose an AHP based resource allocation balance-aware (AESVMP) virtual machine placement strategy, with the help of dynamic virtual machine consolidation technology to dynamically schedule and allocate virtual machine resources, to achieve the goals of energy-saving optimization and reduction in the number of SLAV, ESV and number of virtual machine migrations in the cloud data center. Compared with the benchmark method, AESVMP can optimize the cloud data center of energy consumption, SLAV, ESV, and a number of VM migrations by 27.8%, 92.25% and 94.25% respectively. Compared to the proposed state-of-the-art method LBVMP under the same conditions, the proposed mechanism outperforms in terms of the number of VM migration, SLAV, and ESV.

Nevertheless, there are a few limitations that need to be further addressed in future works. The AESVMP algorithm is slightly less optimized for energy consumption compared to the LBVMP algorithm needs to be further optimized. Also in the future, we will test our approach on real cloud platforms (e.g., Video Cloud Computing Platform and OpenStack) to verify the effectiveness of the AESVMP algorithm in real environments.

Supplemental Information

Supplemental Information 1 Source code.

Click here for additional data file.

The authors would like to thank the anonymous reviewers for their insightful comments and suggestions on improving this article.

Additional Information and Declarations

Competing Interests

Author Contributions

Data Availability

The authors declare that they have no competing interests.

Hangyu Gu conceived and designed the experiments, performed the experiments, analyzed the data, performed the computation work, prepared figures and/or tables, authored or reviewed drafts of the article, and approved the final draft.

Jinjiang Wang conceived and designed the experiments, performed the experiments, performed the computation work, prepared figures and/or tables, authored or reviewed drafts of the article, and approved the final draft.

Junyang Yu conceived and designed the experiments, performed the experiments, performed the computation work, prepared figures and/or tables, authored or reviewed drafts of the article, and approved the final draft.

Dan Wang performed the experiments, performed the computation work, prepared figures and/or tables, authored or reviewed drafts of the article, and approved the final draft.

Bohan Li analyzed the data, authored or reviewed drafts of the article, and approved the final draft.

Xin He analyzed the data, authored or reviewed drafts of the article, and approved the final draft.

Xiang Yin analyzed the data, authored or reviewed drafts of the article, and approved the final draft.

The following information was supplied regarding data availability:

The source code is available in the Supplemental File and at Zenodo with the data: Gu, H. (2023). Cloudsim Cloud Computing Simulation Platform and data set Creators [Data set]. Zenodo. https://doi.org/10.5281/zenodo.10011819.

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
