# Peer review of "Towards virtual machine scheduling research based on multi-decision AHP method in the cloud computing platform"

_PeerJ Computer Science, doi:10.7717/peerj-cs.1675_

## Round 0.1 · accepted · Accept

I really appreciate your interest in the journal. Best.

·

Basic reporting

This paper is the revised version of the paper reviewed by me earlier. This version has addressed the review comments made by me and in its current form paper is reading well and can be accepted. Some minor comments are as follows:

1) Align equations such that the "=" sign between the Left Hand Side and Right Hand Side for all equations are aligned. This will improve readability.
2) From algorithm 1, remove the data types "Double", and "int". We don't need to specify data types in the algorithm.
3) In the abstract instead of (CPU, RAM, BW) it might be better to use (CPU, RAM, NETWORK) or (CPU, RAM, IO). BW (Bandwidth) is a property of the resource and not the resource itself.

Experimental design

The graphs in this version of paper are better and readable.

Validity of the findings

all good.

Additional comments

In its current form paper is acceptable. Reviewers can fix the minor issues mentioned.

·

Basic reporting

The authors have addressed all of my concerns

Experimental design

The authors have addressed all of my concerns

Validity of the findings

The authors have addressed all of my concerns

Additional comments

The authors have addressed all of my concerns